# Evaluating and redesigning the teaching practices of the Kazakh Mythology- engraving techniques

**Karim Baigutov**[1]*, **Aman Ibragimov**[1], **Alzhanov Gadilbek Kh.**[2], **Kumarkan Muratayev**[1], **Taiyrzhan Iskakov**[1], **Dzhanaev Miyat**[1]

**1** Abai Kazakh National Pedagogical University, Almaty, Kazakhstan, **2** Dosmukhamedov Atyrau State University, Atyrau, Kazakhstan

* karimkhan.art@gmail.com

**Data Availability Statement:** The data set related to this study is available on official request as due to ethical concerns we cannot upload the data set on webpages for sharing purposes. In case a

## Abstract

The study was designed to explore and develop art students' interest and engagement in Kazakh mythology and using engraving techniques. The archetypic and challenges of Kazakh mythology in art students were not previously explored. Therefore, the need for study in this domain was essential to cover the gap in the literature. The gap has been recently discovered by Kazakh scholars although it has been explored by foreign authors but the authenticity of such studies remains a question. The study was experimental and the results show a strong association between art's student's engagement with Kazakh mythology using engraving techniques. Moreover, the elective course was selected by 90% class of the art students voluntarily. The teaching method developed for the study provides strong results and the outcome of the experiment was well above 80%. The study used a systematic method along with comparative analysis based on Kazakh myths, culture, ethnology, folklore, mythology, and archeology.

## Introduction

Kazakh mythology is a rich and fascinating area of study that has received increasing attention in recent years [1–4]. Developing students' interest in Kazakh mythology is a challenging task [5, 6]. One possible solution for developing students' interest in Kazakh mythology is the use of engraving techniques. Through the engraving techniques students' interests can be developed as introducing myths and legends to the students has been linked with students' engagement [7]. Engraving is a form of printmaking that involves carving a design into a surface, typically a metal plate or a piece of wood. The design is then inked and printed onto paper, creating a unique and detailed image [8, 9]. Engraving has a long history in Kazakh culture and many of the traditional designs and motifs used in engraving have their roots in Kazakh mythology [2].

The art engraving techniques evolve around the core concept of myth. The concept of myth can be considered among the oldest methods of explaining different social phenomena. Furthermore, "The space of myth created the architecture of the world" [10]. Furthermore, the

scholar wants to work on our data set he/she needs to request on karimbaigutov@gmail.com, or to yerlan_fks@mail.ru Yerlan Seisenbekov - Director Ethical and Data Supervisory Committee Institute of Arts, Culture and Sports Abai Kazakh National Pedagogical University

**Funding:** The author(s) received no specific funding for this work.

**Competing interests:** The authors have declared that no competing interests exist.

transition from archaic myth to mythology led to the formation of many cultural spaces with their mental foundations on which various socio-cultural systems have relied and are based [11]. For example, the association between Kazakh mythology and Turkic mythology [6]. Methodological perspectives of Kazakh art, history, culture, and society as a whole assist in the interpretation of Kazakh myths and mythologies [12]. Until the beginning of the 20[th] century, Kazakh Mythology was not explored separately from folkloristics and only a handful of studies exist in this domain that do not pay attention to the interpretation and analysis of mythical stories [6]. Kazakh myths have an interesting association with stars and the end of the world such as the myth of "Zheti Karakushy" (Big Bear) that consist of seven stars linked with seven thieves; "Temir Kazyk"-the Iron Stake and the two stars near Iron Stake is depicted as two horses tied to the Iron Stake. "Ak Boz At" the white horse; "Kuk Boz At" the blue-gray; and "Akır Zaman" - "The End of the World" [13]. Moreover, various studies have analyzed the art history, artistic, creative, semiotic, and structuralist studies of the mythology of the XX-XXI century [14, 15]. The period emphasizes the universality of the functions of myths, regardless of which "mental environment" they arise in, "traditional and archaic", "Hellenistic", or "bourgeois".

Students' engagement can be obtained by introducing Kazakh mythology as a main course in the curriculum. This may include a brief history of the art engraving techniques, along with an overview of the tools and materials used in engraving, and a demonstration of the engraving process itself. Furthermore, once students have a basic understanding of engraving, they begin to explore the myths and legends of Kazakh culture through the creation of their engravings. Educators may provide students with a selection of traditional Kazakh designs and motifs to use as inspiration for their engravings. Students need to be encouraged to create their designs based on their interpretation of Kazakh mythology. The purpose of the research is to highlight the importance of studying myth and mythology for art students and professionals, as well as to emphasize the relevance of developing critical analysis skills for both traditional and neo-mythological material. The study aims to explore the role of myth and mythological systems in Kazakh history and culture and to familiarize art students with various aspects of Kazakh myth and mythology. The research also emphasizes the need to develop critical thinking skills and the ability to analyze mythological and neo-mythological material in its narrative and artistic expression, which is crucial for future art teachers, graphic artists, and other art professionals.

Mythological consciousness is a process of interpretation as an immanent component of social consciousness [16]. Understanding the "pulsating" nature of mythological consciousness and analyzing its existence in specific socio-historical conditions is an integral part of the professional skills of an art critic [17, 18]. The concept of myth is usually based on "rational findings" and/or "fictional consciousness" Additionally, in the medieval period the concept of myths are often used as a source for the manifestation of power by the elites. In these cases, mythological images, plots, and characters become the object of unlimited manipulations, fixed in the public consciousness by the available mass media. Such processes are especially characteristic of the so-called epochs of "breaking the world" (Weltwende). The term was first introduced into scientific circulation by Arnold Toynbee, who believed, in particular, that the I century BC was a frontier, Weltwende–"breaking of the world", a time when rationalism exhausted all its resources and was replaced by mysticism, stagnation, and apathy [19].

The study of the status of the "mythological" in the mentality of traditional communities can contribute to the explanation of the origins and analogy of the above phenomenon in the modern world, the distinctive feature of which is a real occult boom, the fashion for witchcraft and sorcery. In this light, the current situation is described by researchers as a "postmodern era" or even as a "new Hellenism", a professional art critic. The artist can analyze the evidence of another reanimation of the mythological way of understanding the world–reanimation,

socially-historically conditioned and having obvious historical boundaries [20]. The neo-mythology is a new edition of mythology XIX century, neo-mythology surfaced as a result of the new understanding of the myth and its role in modern society. The myths fall into the domain of the "Quasi-scientific" and through the quasi-scientific approach, the fundamentals of myths can be explored. Additionally, to the greatest extent, quasi-scientific theories touch upon the issues of a society full of the most acute contradictions. Therefore, vulnerable and striving for stability and balance, there is an undeniable shortage of a calm, sober understanding of neo-mythmaking, and the need to develop an adequate worldview attitude to it. The approach requiring researchers to remain in positions of rational criticism can create conditions for achieving a balance of interests of society and the individual based on civil society [21].

This study was designed to explore students' engagement and performance in an experiment designed to promote Kazakh mythology using engraving techniques at the Institute of Arts, Culture, and Sports, Abai University- Kazakhstan.

## The myth–Understanding the origin of myth

The methodological challenges linked with the teaching of Kazakh mythology and students' engagement require an understanding of students' interests and engagement. That is how students can be engaged in learning Kazakh mythology using engraving techniques. (Fig 1) represents the concept of myth in terms of its foundations, historical perspective, and association.

## Methodology

To preserve and develop Kazakh mythology and art, the Institute of Arts, Culture, and Sports, Abai University has conducted an experiment in which a course on "Kazakh mythology and Art" as a subject–as part of the elective discipline (60 credits) was introduced to the art department students (Bachelor and Master students). Mythology is a subject previously taught in Kazakh universities to students of philological, cultural, religious studies, and philosophical, but, unfortunately, not art history or in the specialties of graphics, and painting. There was a need to introduce Kazakh mythology and art as a course that would cover a wide range of problems related not only to the study of various techniques of mythology but also analysis of the main mythologies and familiarization with the main scientific theories in the field of myth

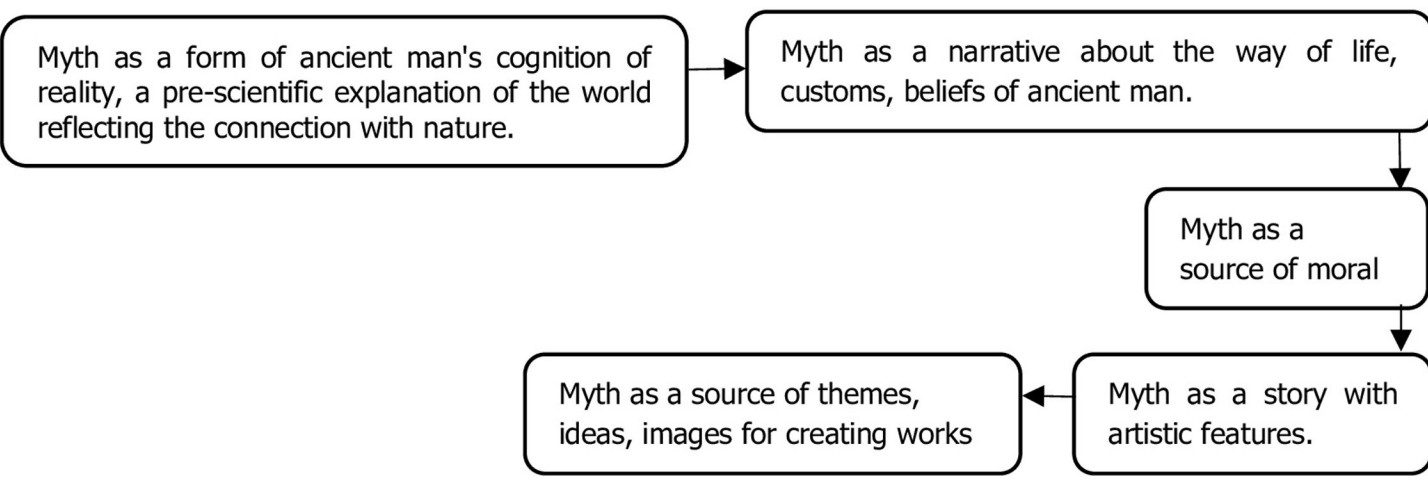

**Fig 1. Understanding the concept of myth.**

and mythology. The purpose of this subject was to gain students' interest and to develop an understanding of Kazakh mythology.

The students' centered research design was adopted to explore students' interest in mythology and art engraving techniques. Furthermore, two study groups were formulated in the study, group 1 students were taught Kazakh mythology through traditional methods. Whereas, group 2 students were taught using the approach we have developed that consists of lectures, seminars, and engraving techniques. The performance of both groups was then compared in terms of their level of knowledge and understanding of Kazakh mythology.

The training course "Kazakh Mythology and Art" consists of 15 lecture topics and 10 seminar topics along with practical classes on engraving techniques from the perspective of Kazakh mythology. The training course begins with "Mythology in art". Additionally, mythology in art course focuses on generating a painting based on myth stories. This allows students to enhance their knowledge and understanding of the history and art of Kazakh mythology along with the Ancient World, and the Middle Ages. The works of various artists, sculptors, and film directors were included in the study. The seminar series on engraving techniques were based on compositional works. The seminar series allows students to participate in the discussion based on each module of training courses.

In addition to studying the main issues of the topics, students were grouped into micro-groups (3 students per group). These micro-groups were assigned engravings and a practical task to prepare art on a particular ritual or various mythical sources (historical or modern) from a particular people or tribe paying specific attention to local ritual with the fusion of the art historians' work, ethnographers, religious scholars, and cultural scientists. The students' outcomes were assessed by the experts and professionals engaged in Kazakh art, history, and archeology. The ethical approval of the study was obtained from the Ethical Research Committee (ERC) of Kazakh National Pedagogical University. Formal written consent was obtained from the participants of the study. Moreover, each participant has explained the purpose and outcomes of the study.

## Theoretical model

(Fig 2) representing the research model used to measure students' engagement.

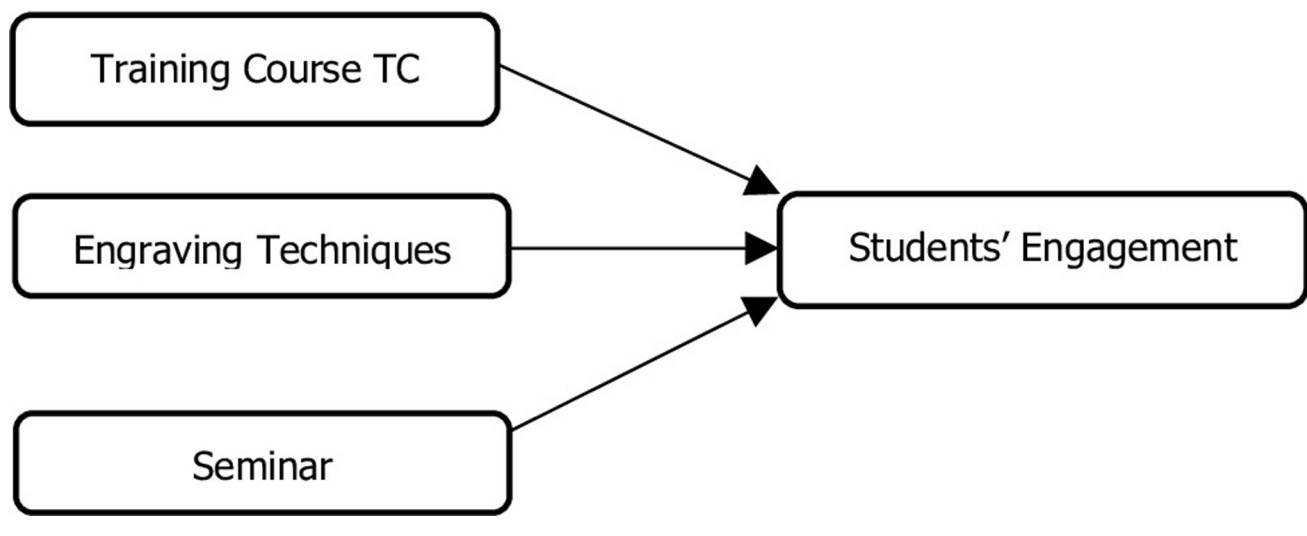

**Fig 2. Theoretical model.**

## Measure

All constructs and items were measured on a Likert five-point scale anchored from 1 (strongly disagree) to 5 (strongly agree).

1. Student engagement: Student engagement was assessed using a 20 items self-developed scale with Cronbach's alpha value (α 0.92). A sample item example: "*The engraving techniques help me in building my interest in Kazakh mythology.*"

2. Training course assessment scale: The training course assessment scale was measured using a 19 items self-developed scale with Cronbach's alpha value (α 0.89). A sample item example: "*The students were able to apply engraving techniques effectively in Kazakh Mythology.*"

3. Seminar assessment scale: Seminar course assessment was measured using a 22 items self-developed scale with Cronbach's alpha value (α 0.88). A sample item example: "*The seminar series helps students develop a more advanced understanding and knowledge of Kazakh mythology."*

4. Engraving technique assessment scale: The engraving technique assessment scale was measured using a 20 items self-developed scale with Cronbach's alpha value (α 0.90). A sample item example: "Engraving techniques help students in innovatively expressing Kazakh Mythology.*"*

## Data analysis and results

Fig 3.

## Insert conceptual model here

**Regression analysis.**   Table 1 provides regression results along with model strength. The model $R^2$ or the coefficient of determination shows 93.9% of variance in the dependent variable (Students' engagement) that is explained by the independent variables (training course, engraving techniques, and seminar). The association between TC and SE was statistically

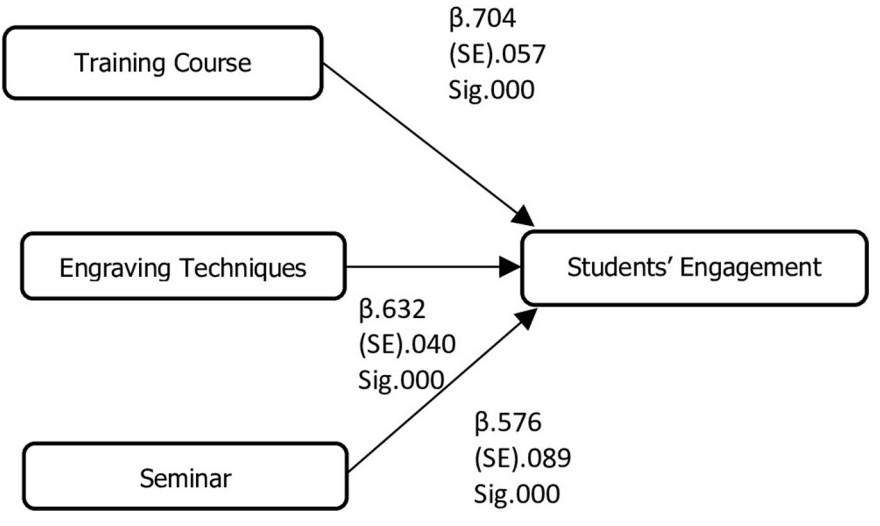

**Fig 3. Conceptual model 1.**

**Table 1. Regression model.**

| Paths | Estimate | Standard Error | f-value | Mean Square | p-value | R Square |
|---|---|---|---|---|---|---|
| TC→SE | .704*** | .057 | 117.644 | 14.055 | 0.000 | .939 |
| ET→SE | .632*** | .040 | | | | 93.9% |
| SM→SE | .576*** | .089 | | | | |

significant ($\beta$ = .704; p = 0.000) with students' engagement; moreover, the association between engraving techniques and students' engagement was also statistically significant ($\beta$ = .632; p = 0.000) and the association between seminar series and students' engagement was also statistically significant ($\beta$ = .576; p = 0.000). (Significance levels: ***p< .001; *p< .05; SMST = TC = Training Course; ET = Engraving Techniques; SM = Seminar; Students' Engagement SE.)

The regression results show the effectiveness of the experiment, the positive association between variables shows that students felt motivated and positively get engaged with Kazakh mythology through the course and seminar that we introduced. This shows the effectiveness of course contents and seminar series. Moreover, the engraving technique's positive association shows that it can deliver effective results in developing and preserving Kazakh mythology. Kazakh mythology which was previously ignored in arts discipline needs to be taught in the art department across the country to promote and preserve cultural and historical values.

**Pearson correlation.** Table 2 shows the correlations analysis, ET has a moderate positive and statistically significant (p .000) association with TC (r = .546**). SM has a moderate positive association with TC (r = .567**) and with ET (r = .667**). Furthermore, SE has a moderate positive association with TC (.471**), ET (.595**), and SM (. 665**). Therefore, from the correlation analysis, we explore that all variables are positively associated. (*. Correlation is significant at the 0.05 level (2-tailed); **. Correlation is significant at the 0.01 level (2-tailed).

**Comparison.** Table 3 shows the comparison between group 1 and group 2, furthermore, it shows the mean scores difference of five indicators. The negative difference between mean scores shows the effectiveness of the integration method that we have developed as compared to the traditional theory-based method that does not support practicality. The Group 1 students were taught Kazakh mythology through traditional methods (Lectures and assignments), whereas, group 2 students were taught using the approach we have developed that consists of lecturers (using visual aids), seminars (students and experts), and engraving techniques.

**Table 2. Correlation analysis.**

| **Correlations** | | | | | |
|---|---|---|---|---|---|
| | | TC | ET | SM | SE |
| TC | Pearson Correlation | 1 | | | |
| | Sig. (2-tailed) | | | | |
| | N | 46 | | | |
| ET | Pearson Correlation | .546** | 1 | | |
| | Sig. (2-tailed) | .000 | | | |
| | N | 46 | 46 | | |
| SM | Pearson Correlation | .567** | .667** | 1 | |
| | Sig. (2-tailed) | .000 | .000 | | |
| | N | 46 | 46 | 46 | |
| SE | Pearson Correlation | .471** | .595** | .665 | 1 |
| | Sig. (2-tailed) | .000 | .000 | .000 | |
| | N | 46 | 46 | 46 | 46 |

**Table 3. Kazakh mythology.**

| Items | Traditional | Integration | Mean Difference MD = T-I | Significant |
|---|---|---|---|---|
| | | | | |
| Students' engagement | 2.1342 | 6.4323 | -4.2981 | <0.01 |
| Knowledge acquisition | 3.8734 | 5.3532 | -1.4798 | <0.01 |
| Kazakh Mythology Understanding | 1.9173 | 5.2487 | -3.3314 | <0.01 |
| Course attendance | 3.6131 | 6.2419 | -2.6288 | <0.01 |
| Students performance | 2.2421 | 5.7548 | -3.5127 | <0.01 |

**Findings.** The results of the study showed that group 2 which was taught Kazakh mythology through the art of engraving performed significantly better in terms of knowledge acquisition and understanding of Kazakh mythology. As compared to the group that was taught Kazakh mythology using traditional methods. This suggests that the use of art and visual aids in teaching mythology can be a highly effective method for improving student learning outcomes. The study also revealed that students found the use of art and engraving to be a more engaging and interesting way to learn about mythology. Students were able to better remember and retain the information that they received through visual aids and artistic depictions. The findings of the study suggested that the use of art and visual aids in teaching mythology can be a highly effective way to improve student learning outcomes and engagement. It also highlights the importance of incorporating a variety of teaching methods and approaches to help students learn and retain information. The incorporation of lectures, seminars, with engraving techniques is directly associated with the success of mastering skills and students' engagement.

## Discussion

The mythology of the Kazakh people and practice in the art of engraving is the topic of the "Mythopoetic model of engraving" the process of studying which involves particular drawing sketch designs and the transition to engraving technology dedicated to the reflection of one of the variants of the model of graphic techniques but also the theories associated with Kazakh myth, culture, and folklore as well as sculpture and traditional costume. Although the degree of independence in performing this task is usually much greater because students of Kazakh mythology at the beginning of the week find it difficult to perform and get engaged however, once students begin to develop an understanding the level of engagement increases and the difficulty level decreased significantly. The research model was driven by theory, history, art, mythology, practice, and engraving techniques.

Mythology in engraving was divided into several extensive practical classes so that students progress with the subject in stages thus reducing the level of difficulty. The categories were divided into scientific literature that included the works of art historians, philosophers, religious scholars, linguists, folklorists, sociologists, ethnographers, and anthropologists. Furthermore, due to the limited existence of educational materials and anthologies that would cover the whole range of necessary issues regarding Kazakh mythology was still missing from the Kazakh language, and recently work on this domain started. S. Kondybay's four-volume research work "Argy-Kazakh Mythology" which is considered an "encyclopedic treatise" is extremely valuable for students in understanding Kazakh mythology (Kondybai, 2004). The curriculum on Kazakh mythology consists of fundamentals of mythology, modern engraving techniques, motifs, symbols, demonology, and mythological characters. Furthermore, as a theoretical basis of current research, in addition to scientific and popular scientific works, students were encouraged to use engraving techniques such as linocuts, woodcuts, and

lithographs to develop their understanding of Kazakh mythology. The association between engraving techniques and mythology can be found throughout the history of art.

The development of characterization of Kazakh mythology is particularly popular among students, through which they can express their creative abilities. When compiling the characteristics of Kazakh mythology, it is recommended to rely on the framework proposed by Lyudmila Nikolaevna Vinogradova (Vinogradova, 2000:60–67). The purposed framework in our opinion, contains the optimal set of positions for a comprehensive description of mythological characters. Furthermore, students that want to improve their academic performance are instructed to compile thematic glossaries on individual sections of the discipline, annotated reviews of internet resources on mythology and essays on a predetermined topic, and analytical reviews of popular science and feature films that address the corresponding theme and the final result of the picture made by the technique of engraving.

## Future recommendation and conclusion

The association between mythology and the art of the ancient world would be a valuable addition to the literature. It is also advisable to develop teaching techniques using technology and practical classes using the various technique of art. The design of course content linking history, the art of primitive society, and the proto-Kazakh mythology of the ancient people of the Saks can result in better student engagement. The expression of ideas through the use of themes and symbolism present in Kazakh mythology will help in the development of the field. Through the art of engraving as a means of teaching Kazakh mythology, educators can engage students in a creative and interactive learning experience that connects them to the rich cultural heritage of Kazakhstan. This approach not only helps students to develop a deeper understanding and appreciation of Kazakh mythology but also fosters important skills such as creativity, critical thinking, and cultural awareness. Furthermore, engraving not only allows students to learn about Kazakh mythology through a hands-on and visually engaging medium, but it also provides a bridge between the past and the present. By studying traditional Kazakh engraving techniques and motifs, students can gain a deeper understanding of the cultural heritage of Kazakhstan, while also exploring how this heritage can be reinterpreted and applied in new and creative ways.

In addition to traditional engraving techniques, modern technology can also be used to enhance the teaching of Kazakh mythology. For example, digital engraving software can be used to create and manipulate designs, while laser engraving machines can be used to produce high-quality engravings quickly and efficiently. By incorporating modern technology into the teaching of Kazakh mythology, educators can help students to develop important technical skills that are relevant to the modern world, such as digital design, computer-aided manufacturing, and graphic design. Students can continue to explore and appreciate the rich cultural heritage of Kazakhstan through the creation of engravings that reflect their unique interpretations and perspectives. Overall, the methodology and technology of teaching Kazakh mythology through the art of engraving provide a powerful and engaging way to connect students with the cultural heritage of Kazakhstan, while also fostering important skills and abilities that are relevant to the modern world. Whether teaching in a formal classroom setting or as a community-based project, this approach to teaching Kazakh mythology has the potential to inspire creativity, cultural awareness, and a deeper understanding of the world. Knowledge of the specific features of the myth, the main mythological plots, and characters become a support for the study of folklore and some of the creative works of famous artists. Among the reasons for turning to mythology in the analysis of works of art, along with deepening penetration into the text, and in working with paintings, students need to master the

individual creative approach of the artist, art historian to mythological material as one of the features of his work. However, the problem of involving mythology as a factor in deepening the analysis of artistic creativity is currently insufficiently developed. Thus, as a result of studying the discipline "Kazakh mythology and Art", students of art historians, graphic artists, and pedagogical specialties manage to form a fairly holistic view of myth and mythology as one of the oldest cultural and historical phenomena and the sphere of activity of creative consciousness and develop skills of critical comprehension of mythological material in all its manifestations.

## Author Contributions

**Conceptualization:** Karim Baigutov, Aman Ibragimov.

**Data curation:** Karim Baigutov, Alzhanov Gadilbek Kh., Kumarkan Muratayev.

**Formal analysis:** Kumarkan Muratayev, Taiyrzhan Iskakov.

**Funding acquisition:** Alzhanov Gadilbek Kh.

**Investigation:** Karim Baigutov.

**Methodology:** Karim Baigutov, Dzhanaev Miyat.

**Resources:** Dzhanaev Miyat.

**Software:** Taiyrzhan Iskakov.

**Visualization:** Taiyrzhan Iskakov.

**Writing – original draft:** Karim Baigutov.

**Writing – review & editing:** Karim Baigutov.

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
