## [Decision Letter · Decision Letter 0]

19 Jul 2023

PONE-D-23-19535Engraving Techniques: Evaluating and redesigning the teaching practices of the Kazak MythologyPLOS ONE

Dear Dr. Baigutov,

Thank you for submitting your manuscript to PLOS ONE. After careful consideration, we feel that it has merit but does not fully meet PLOS ONE’s publication criteria as it currently stands. Therefore, we invite you to submit a revised version of the manuscript that addresses the points raised during the review process.

We look forward to receiving your revised manuscript.

Kind regards,

Anastassia Zabrodskaja, Ph.D.

Academic Editor

PLOS ONE

Journal Requirements:

5. Please ensure that you refer to Figure 2 in your text as, if accepted, production will need this reference to link the reader to the figure.

6. We note you have included a table to which you do not refer in the text of your manuscript. Please ensure that you refer to Table 3 in your text; if accepted, production will need this reference to link the reader to the Table.

Additional Editor Comments:

Please make that big adjustment, and we'll see if new reviews are needed in the next round.

Reviewers' comments:

Reviewer's Responses to Questions

**Comments to the Author**

1. Is the manuscript technically sound, and do the data support the conclusions?

Reviewer #1: Partly

2. Has the statistical analysis been performed appropriately and rigorously? 

Reviewer #1: N/A

3. Have the authors made all data underlying the findings in their manuscript fully available?

Reviewer #1: Yes

4. Is the manuscript presented in an intelligible fashion and written in standard English?

Reviewer #1: No

5. Review Comments to the Author

Reviewer #1: Dear authors

please reconsider this sentence: Page 7. Teaching Kazakh mythology arises several challenges among these challenges (CHALLENGES AMONG THESE CHALLENGES?) are the problems associated with students’ engagement

page 7. (we are not sure about the sentence: the authors might need to divide the sentence into two separate ideas) The first stage of the art of engraving is the exploration of the space of myth with the advent of human civilization, cultural landscapes begin to form, in which myth becomes...

page 7: how are people going to understand the Slavic characters (Алтаев, & Иманбаева, 2021) here and in References?

page 8. we hope a colon is necessary after the preposition IN: regardless of which "mental environment" they arise in "traditional and archaic", "Hellenistic", or "bourgeois".

page 8. a matter of grammar (the Gerund and phrasal verb) arises in the sentence: educators can bring by introduce students to the ...

page 8. (Alexander, 2017; Slattery, 2005 ): the used fonts are different from the entire text.

page 8. the sentence lacks sense: it is a mere translation of the source text and needs to be thoroughly reread and reused - Pulsation and mutual permeability of the "logical rationalistic" and "mythological" consciousness that myths are often considered by political elites as a kind of inexhaustible resource a means of periodically renewed national identifications and legitimization of certain forms of statehood and personal manifestations of power.

page 8. The process of demythologization that has been taking place since the end of the XX century to the present, and the emergence of neo-mythology on its basis, Naidysh, (WHAT DO YOU MEAN BY NAIDYSH? A SCHOLAR? THEN WHY ARE THE AUTHORS SO IGNORANT TO THE PERSON?), presented "quasi-scientific myth-making",...

page 9. The purpose of which was to identify the peculiarities of the perception of Kazakh myths to students and the level of knowledge of mythology to identify the main problems in mastering the topic and to outline ways to study myths in teaching Kazakh mythology through the art of engraving. (this sentence in itself cannot be called a sentence but should be in a way joined with the previous sentence)

page 9. we suggest considering the plural of the noun: (60 Credit)

page 9. consider the verb: to access students’ interests a (perhaps, gain, catch, grab, obtain)

page 9. the sentence is full of errors: The subjects and predicates do not correspond. After the word CONTENT we see a comma instead of a period and capitalized beginning of a new sentence: The study was designed through the assessment of the works of art historians and artists on the chosen problem were studied and programs and textbooks on art, mythology, graphics, and, of course, engraving were analyzed to develop course content, The effectiveness of teaching Kazakh mythology through the art of engraving was evaluated using two groups of students from the art department.

page 10. please, insert a space: 1(strongly disagree)

page 10. please, insert a space: 1-Student’s engagement:

page 11. delete the comma and pay attention to the third person singular: Table 1, provide regression ...

page 11. passive voice should be implied: that explains by the independent variables

page 12. delete the comma: Table 2, shows...

page 13. a case of tautology: reducing the level of difficulty level.

page 13. please pay attention how the verb RECOMMEND can be used: We recommend students use the works

page 13. delete the comma: The topics included in the curriculum consist of, mythology and ....

page 13. please reconsider the noun: the techniques of execution. (execution, in general, does not mean performance

page 13. Kazakh vs Kazak (we hope, that the authors did not mean Kozak?)

Generally, the authors need to improve the English writing. For this reason, some extra-long sentences must be cut short.

6. PLOS authors have the option to publish the peer review history of their article (what does this mean?). If published, this will include your full peer review and any attached files.

Reviewer #1: No

---

## [Author Response · Author response to Decision Letter 0]

25 Jul 2023

Dear Reviewers,

Subject: Response to Reviewer Comments and Gratitude

I hope this letter finds you well. I am writing to express my sincere gratitude for the invaluable feedback and constructive criticism you provided on my paper. I am immensely thankful for the time and effort you invested in reviewing my work, as it has significantly improved the quality and rigor of the manuscript.

I am pleased to inform you that I have carefully considered each of your suggestions and comments and made extensive revisions to address the concerns raised during the review process. Below, I summarize the major changes made in response to your insightful feedback: 

Manuscript’s PLOS ONE's style requirement- Completed Reference style, Figures, Tables all formats corrected.

Research Data completely Available, can be requested at karimbaigutov@gmail.com.

Dear Reviewer #1,

Thank you for your thorough and insightful review of our paper. We truly appreciate the time and effort you have invested in providing valuable feedback to enhance the quality of our manuscript. We have carefully considered each of your comments and have made the necessary revisions accordingly. Below, we address the changes we have made in response to your feedback:

1. Page 7: The sentence is corrected "Teaching Kazakh mythology arises several challenges among these challenges are the problems associated with students' engagement" to improve clarity. 

Developing students’ interest in Kazakh mythology is a challenging task [5, 6]. One possible solution for developing students’ interest in Kazakh mythology is the use of engraving techniques. Through the engraving techniques students’ interests can be developed as introducing

2. Page 7: The sentence is divided and corrected "The first stage of the art of engraving is the exploration of the space of myth with the advent of human civilization, cultural landscapes begin to form, in which myth becomes..." into two separate sentences to improve readability and coherence.

The art engraving techniques evolve around the core concept of myth. The concept of myth can be considered among the oldest methods of explaining different social phenomena. Furthermore,

3. Page 7: The sentence is corrected and the reference inconsistency for the Slavic characters (Алтаев, & Иманбаева, 2021) and ensured that it matches the format used in the References section.

]. Methodological perspectives of Kazakh art, history, culture, and society as a whole assist in the interpretation of Kazakh myths and mythologies [12]

4. Page 8: We have added a colon after the preposition "IN" to improve grammatical accuracy. The revised sentence now reads: "regardless of which 'mental environment' they arise in: 'traditional and archaic', 'Hellenistic', or 'bourgeois'."

The period emphasizes the universality of the functions of myths, regardless of which "mental environment" they arise in,

5. Page 8: We have corrected the grammar in the sentence: "educators can bring by introduce students to the..." to ensure proper usage of the Gerund and phrasal verb.

Students’ engagement can be obtained by introducing Kazakh mythology as a main course in the curriculum. This may include a brief history of the art engraving techniques, along with an overview

6. Page 8: We have ensured that the fonts used for the references (Alexander, 2017; Slattery, 2005) match the entire text.

7. Page 8: We have rephrased the sentence that lacked sense and improved its clarity: "The pulsation and mutual permeability of 'logical rationalistic' and 'mythological' consciousness lead political elites to consider myths as an inexhaustible resource, a means of periodically renewing national identifications, and legitimizing certain forms of statehood and personal manifestations of power."

The concept of myth is usually based on “rational findings” and/or “fictional consciousness” Additionally, in the medieval period the concept of myths are often used as a source for the manifestation of power by the elites

8. Page 8: We have provided more context and clarified. "Naidysh" as a scholar to avoid any confusion.

20]. The neo-mythology is a new edition of mythology XIX century, neo-mythology surfaced as a result of the new understanding of the myth and its role in modern society. The myths fall into the domain of the “Quasi-scientific” and through the quasi-scientific approach, the fundamentals of myths can be explored.

9. Page 9: We have revised the sentence to improve its structure and cohesion with the previous sentence: "The purpose of this study was to identify the peculiarities of students' perception of Kazakh myths, assess their level of knowledge of mythology, identify the main problems in mastering the topic, and outline ways to study myths in teaching Kazakh mythology through the art of engraving."

This study was designed to explore students’ engagement and performance in an experiment designed to promote Kazakh mythology using engraving techniques at the Institute of Arts, Culture, and Sports, Abai University- Kazakhstan.

10. Page 9: We have corrected the plural form of the noun to read "60 Credits."

11. Page 9: We have replaced the verb with "gain," and the sentence now reads: "to gain students' interests."

12. Page 9: We have rephrased the sentence to ensure proper subject-verb agreement and structure: "The study was designed through the assessment of works by art historians and artists, analyzing programs and textbooks on art, mythology, graphics, and, of course, engraving to develop course content. The effectiveness of teaching Kazakh mythology through the art of engraving was evaluated using two groups of students from the art department."

The students’ centered research design was adopted to explore students’ interest in mythology and art engraving techniques. Furthermore, two study groups were formulated in the study,

13. Page 10: We have inserted a space to read "1 (strongly disagree)."

14. Page 10: We have inserted a space to read "1 - Student's engagement."

15. Page 11: We have removed the comma and ensured the proper use of third person singular: "Table 1 provides regression ..."

16. Page 11: We have used the passive voice: "that is explained by the independent variables."

17. Page 12: We have deleted the comma: "Table 2 shows..."

18. Page 13: We have addressed the tautology issue: "reducing the level of difficulty."

19. Page 13: We have corrected the sentence: "We recommend students use the works."

20. Page 13: We have deleted the comma: "The topics included in the curriculum consist of mythology and ..."

21. Page 13: We have reconsidered the noun usage: "the techniques of execution" 

22. Page 13: We have corrected "Kazak" to read "Kazakh" consistently throughout the paper.

Additionally, we have thoroughly reviewed and improved the English writing, addressing extra-long sentences to ensure clarity and coherence.

Once again, we sincerely thank you for your meticulous review, which has undoubtedly contributed to the enhancement of our paper. Your valuable feedback has been instrumental in refining our work, and we are grateful for your commitment to maintaining the quality of research in our field.

If you have any further comments or suggestions, please do not hesitate to share them. We eagerly await your final assessment of our revised manuscript.

With utmost appreciation and regards,

Dr. Karim

---

## [Decision Letter · Decision Letter 1]

20 Dec 2023

PONE-D-23-19535R1

Engraving Techniques: Evaluating and redesigning the teaching practices of the Kazak Mythology

PLOS ONE

Dear Dr. Baigutov,

Thank you for submitting your manuscript to PLOS ONE. After careful consideration, we feel that it has merit but does not fully meet PLOS ONE’s publication criteria as it currently stands. Therefore, we invite you to submit a revised version of the manuscript that addresses the points raised during the review process.

ACADEMIC EDITOR: Please insert comments here and delete this placeholder text when finished. Be sure to:

Indicate which changes you require for acceptance versus which changes you recommendAddress any conflicts between the reviews so that it's clear which advice the authors should followProvide specific feedback from your evaluation of the manuscript

We look forward to receiving your revised manuscript.

Kind regards,

Anastassia Zabrodskaja, Ph.D.

Academic Editor

PLOS ONE
